# Reflective Agents for Knowledge Graph Traversal

Michal Chudoba*[1]

[1]University of Stavanger, Norway
[1]michal.chudoba@uis.no

## Abstract

Current research on Retrieval Augmented Generation (RAG) for Knowledge Graphs often relies on graph pruning to manage the scale of the data. This approach is not feasible for dense, highly structured environments like rigid ontologies, where every node has significant interconnected value. The sheer size of these graphs inhibits the effectiveness of standard semantic retrieval methods. To overcome this limitation, we introduce a novel approach using an autonomous agent that dynamically traverses the graph to retrieve information. A key contribution of our work is the integration of a feedback mechanism that informs the agent about its general performance and specific tool utilization, thereby enhancing its traversal efficiency. We validate our method through a systematic study on ontologies of varying sizes, employing a user simulator to generate realistic tasks for knowledge graph construction and querying. Our findings demonstrate the current problems with information retrieval in large, non prunable knowledge structures.

## 1 Introduction

Large Language Models (LLMs) [1] have demonstrated remarkable capabilities in natural language understanding and generation. However, their reliance on internal knowledge often leads to factual inaccuracies or hallucinations when confronted with topics outside of their domain [2]. Retrieval Augmented Generation [3] was developed to address this by providing LLMs with external knowledge sources, giving relevant context to construct more accurate responses. Despite these advancements, standard RAG systems face significant challenges. When applied to specialized domains, generic retrieval methods may fail to capture nuanced details within documents, leading to suboptimal context retrieval [4]. A similar problem occurs with scale, as the number of documents grows, identifying the correct information becomes increasingly difficult for dense retrievals that are employed [5]. In more structured domains, such as scientific literature or legal precedent, conventional RAG does not account for the critical relationships between documents, such

as citations or dependencies, resulting in incomplete answers [6]. To mitigate these issues, several advanced frameworks have emerged. Self-RAG [7] improves the quality of the response through reflection and by generating different responses for each retrieved document. GraphRAG [8] [9] addresses the issue of interconnectedness by using a graph-like structural representation of the source documents, enabling retrieval based on both semantic and structural similarity. Meanwhile, frameworks such as LongRAG [10] leverage the expanding context windows of modern LLMs to process larger chunks of text, shifting more of the extraction burden onto the model itself. Although effective, these methods share a common prerequisite: the need to pre-process and embed the entire knowledge base, an inefficient and often impractical task for massive, dynamic datasets. An alternative paradigm that bypasses the limitations of static retrieval is the use of autonomous agents [11]. Pioneering work like WebGPT [12] demonstrated the power of equipping LLMs with tools, such as a search engine, to actively seek out information. The recent development of model protocols (MCP) [13] further promises to allow models to request knowledge precisely when needed, optimizing the retrieval process. In this paper, we extend the autonomous agent paradigm to the domain of large-scale structured knowledge graphs. We argue that for complex graph structures, a dynamic traversal agent offers a more effective and scalable alternative to semantic-based retrieval performed by traditional RAG systems. The agent can navigate the graph's topology, follow relational paths, and iteratively build context in a manner that is impossible for embedding-based search. We explore how an agent utilizes its tools for graph navigation and how its behavior is refined through the introduction of a reflective loop. Our primary contributions are threefold:

- An investigation of the tool utilization patterns of an autonomous agent operating within complex graph structures.

- The design and implementation of a self-reflection loop that provides targeted feedback to enhance the agent's tool-use efficiency and traversal strategy.

- Performance analysis of our agent across knowledge graphs of varying domains and sizes,

---

*Corresponding Author.

Proceedings of the 7th Northern Lights Deep Learning Conference (NLDL), PMLR 307, 2026.

demonstrating the viability and scalability of our approach.

This paper is structured as follows. Section 2 reviews the related literature. Section 3 details our proposed agent-based traversal approach. Section 4 presents the experimental results of our evaluation on selected ontologies and knowledge graphs. Section 5 then discusses the importance of tool selection with the ablation study.

# 2 Related work

The evolution of RAG has been marked by a continuous effort to enhance the quality and relevance of the retrieved context, especially when dealing with structured data. This progression can be broadly categorized into advancements in graph-based RAG, the integration of reflective mechanisms, and the rise of autonomous agents for dynamic information seeking.

## 2.1 From Standard to Graph-Based RAG

Conventional RAG systems, while effective for unstructured text, often fail to capitalize on the inherent relationships within structured knowledge bases [6]. To address this, GraphRAG [8] has emerged as a prominent paradigm that represents knowledge as a graph, allowing retrieval based on both semantic content and structural links [9]. The GraphRAG framework typically involves sophisticated query processing (e.g. entity recognition, relation extraction), a retrieval phase that can employ methods from heuristic-based graph traversal to GNN-based embeddings, and an organization step like graph pruning to refine the context before generation. Papers like OntoRAG [14] demonstrate the power of this approach by automatically constructing an ontology from domain-specific documents and leveraging the created structure to facilitate more effective multi-hop reasoning . Similarly, other works have used knowledge graphs to find thematic similarities in specialized domains such as legal judgments by representing documents as nodes with feature vectors based on rhetorical roles [15]. However, these methods often rely on a "retrieve-then-generate" workflow, which still requires processing and embedding a significant portion of the graph, a process that becomes inefficient at a large scale, especially for dense knowledge structures where aggressive pruning is not viable.

## 2.2 Agent-based Graph Traversal

An alternative to static retrieval is the use of autonomous agents. These agents can interact with knowledge sources dynamically, iteratively building a context. This approach has its foundations in work like WebGPT [12], which equipped an LLM with a search engine. This concept has been extended to structured data, with research on Knowledge Graph Prompting [16] demonstrating an LLM agent that traverses a graph by generating prompts at each step to determine its next move. This method acknowledges the importance of graph density and shows that an agent's path-finding ability is critical for gathering supporting facts. Our work builds directly on this agent-based traversal paradigm. However, where previous work has focused on the feasibility of traversal itself, we concentrate on the agent's tool-use patterns and the introduction of a feedback mechanism to optimize its navigation strategy, which has not been the primary focus of prior graph traversal agents.

## 2.3 Self-Reflection and Reflection Loops

The limitations of static retrieval strategies have motivated research into adaptive RAG systems that can refine their approach based on retrieval quality and task performance. Self-RAG [7] represents a significant advancement in this direction, introducing a reflective component where the model assesses the relevance of retrieved passages and assesses its own generated output, making dynamic decisions about whether additional retrieval is necessary. Reflexion [17] employs a verbal reinforcement learning framework where an Actor agent's performance is evaluated by a separate Evaluator, while a Self-Reflection model generates explicit textual feedback to guide subsequent attempts. This shows that structured feedback mechanisms can significantly improve agent performance on complex reasoning tasks that require iterative refinement. In the context of knowledge graphs, refinement loops have been applied to improve the quality of retrieved RDF triplets, creating a feedback cycle that improves the semantic precision of retrieved structured data [6].

Although these approaches have shown promise in improving retrieval quality and reasoning performance, they primarily focus on content relevance rather than navigation efficiency in structured knowledge spaces. The feedback mechanisms in Self-RAG are designed to assess document relevance, and Reflexion's framework, though powerful, has not been specifically adapted for optimizing traversal strategies.

In summary, while previous work has independently advanced graph-based RAG, agentic traversal, and reflective mechanisms, our research

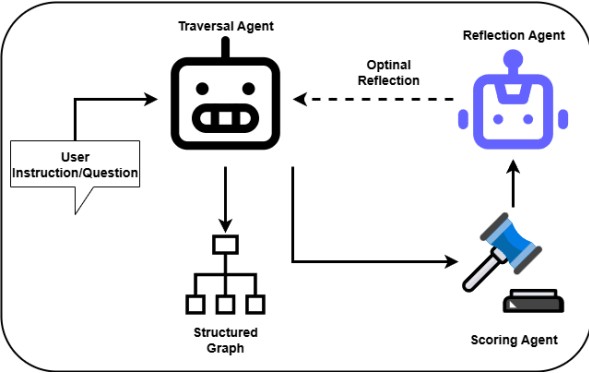

**Figure 1.** Methodology of the Traversal Agent with Reflection. The Agent starts with user query and uses his tools to navigate the underlying graph. After finishing the traversal, nodes the agent marked as important are retrieved and scoring agent judges the retrieval performance (Precision and Recall, with Recall weighted more). Reflection Agent ponders over the score and the message trace generated by the Traversal Agent and generated possible recommendation for the next iteration.

provides a novel synthesis of all three. We differentiate ourselves by implementing a reflection loop not for critiquing the retrieved content, as in Self-RAG, but for optimizing the process of traversal itself. By applying this principle to the specific challenge of navigating dense ontologies, we address a critical efficiency and scalability gap left by existing agent and graph retrieval frameworks.

## 3 Methodology

While the general methodology overview can be found in Figure 1, in this section we would like to dive deeper into the individual parts as presented in the Figure.

### 3.1 Traversal Agent

The traversal agent was inspired by the implementation of Multi-Document QA with Knowledge Graphs [16], but instead of the generation of the next paragraph, we have introduced a set of traversal and matching tools for the agent to use. We have deemed that some matching tool is necessary to allow the agent to obtain the initial node from which to traverse. For traversal tools, following where given to the agent:

- get_neighbors - This tool allows the Agent to get close neighbors (objects and subject connected with some predicate) of any specified node.

- move_to - Agent movement tool, moving to any node will reveal nodes properties, which are otherwise not given for brevity.

- mark_nodes - Agents retrieval tool inspired by WebGPT. Agent marks noes while giving his confidence and reasoning how the node is relevant to the question/instruction.

- get_predicates_for_entity - This tool allows the agent to obtain predicates associated with any given entity, allowing the model to determine the need to explore the node further.

- end_traversal - Tool for the agent to signal the end of his traversal.

During inference, agent is limited to a certain amount of move and mark tool calls, while all calls are monitored for count. An example prompt for this agent is mentioned in the Appendix A

### 3.2 Scoring Agent

The scoring agent tries to score how well the agent accomplishes his goal. The agent sees the initial user question/query and the marked nodes. Based on that, it scores the traversal retrieval in 2 areas. 1. is answer recall, which scores if the marked nodes seem to answer or grasp the intent of the user. 2. is the precision, which scores if there are no redundant nodes found in between the answers. Because the intent of the traversal agent is to recall, we have instructed the scoring agent to weight the recall part more. The exact prompt for the scoring agent can be found in Appendix A

### 3.3 Reflection Agent

The reflection agent tries to generate insight that may help the Traversal Agent in the next iteration. However, compared to the reflection present in Self-Rag and Reflexion, our reflection agent does not see the initial user input. Instead, only the trace of the Traversal Agent and the Score of the scoring agent are provided. This is to guide it, together with our prompt (See Appendix A), to reflect more on tool utilization rather than question itself. In our hypothesis, this should result in the next traversal being more effective.

## 4 Experiments

For our benchmark datasets, we selected a range of ontologies to test our agent under different conditions. We chose the Drilling and Wells Interoperability Standards (D-WIS) ontology for its complexity, containing over 6,000 classes with many having similar semantics (e.g. *CompressiveDrillPipe* vs. *DrillPipe*), which demands nuanced reasoning. To broaden our evaluation, we included the Geolink Base Ontology (GBO) [18], another large-scale domain-specific graph, and the simple and

**Table 1.** Recall Results. The Traversal Agent consistently achieves high recall, with reflection providing a boost on more complex tasks.

| Model | D-WIS | FoF | GBO | MetaQA | PQL |
|---|---|---|---|---|---|
| Dense | 21.4 % | 62.3 % | 70.8 % | - | - |
| Dense with ER | 84.7 % | 78.0 % | 83.0 % | - | - |
| SPARQL | 85.4 % | 84.2 % | 94.7 % | - | - |
| N-Hop | - | - | - | 88.7 % | 69.0 % |
| Traversal w/o Ref | 90.8 % | **89.7 %** | **95.5 %** | 71.5 % | 85.4 % |
| Traversal w/ Ref | **92.5 %** | **89.7 %** | 93.2 % | 84.3 % | **88.1 %** |

**Table 2.** Precision Results. Reflection generally improves precision, except on simple graphs where initial performance is already high.

| Model | D-WIS | FoF | GBO | MetaQA | PQL |
|---|---|---|---|---|---|
| SPARQL | 46.7 % | **68.7 %** | 70.8 % | - | - |
| N-Hop | - | - | - | 47.6 % | 48.2 % |
| Traversal w/o Ref | 68.4 % | 65.0 % | 69.5 % | 51.3 % | 59.7 % |
| Traversal w/ Ref | **71.2 %** | 64.2 % | **75.9 %** | **59.8 %** | **62.0 %** |

**Table 3.** Average number of tool calls per task. Reflection guides the agent toward more efficient exploration on complex graphs but can lead to over-exploration on simple ones.

| Model | D-WIS | FoF | GBO | MetaQA | PQL |
|---|---|---|---|---|---|
| **w/o Reflection** | | | | | |
| get_best_matches | 1.6 | 1.3 | 1.4 | 1.8 | 1.2 |
| get_neighbors | 3.2 | 1.5 | 6.1 | 12.0 | 4.4 |
| mark_nodes | 4.2 | 4.8 | 4.8 | 3.8 | 2.2 |
| move_to | 4.0 | 1.8 | 3.3 | 4.0 | 3.8 |
| get_predicates_for_entity | 2.0 | 0.3 | 0.3 | 4.2 | 1.8 |
| **w/ Reflection** | | | | | |
| get_best_matches | 1.0 | 2.0 | 1.2 | 1.4 | 1.2 |
| get_neighbors | 5.2 | 3.8 | 3.6 | 9.4 | 3.8 |
| mark_nodes | 3.4 | 3.6 | 4.2 | 3.6 | 2.2 |
| move_to | 4.6 | 2.7 | 3.4 | 5.6 | 2.4 |
| get_predicates_for_entity | 0.6 | 1.3 | 0.5 | 3.6 | 2.6 |

well-known Friends of a Friends (FoF) [19] ontology to validate performance on less complex structures. As there is no instruction set associated with these ontologies, we used a User Simulator to generate tasks aimed at constructing and querying a Knowledge Graph from the ontology (see Appendix A). To explore if our findings generalize, we also used the MetaQA [20] and PathQuestion-Large (PQL) [21] datasets, which provide established knowledge graphs and associated questions.

For our baselines on the ontology datasets (D-WIS, GBO, FoF), we employed standard Dense Retrieval, a variant with an Entity Recognition (ER) preproccessing, and a SPARQL-generating Agent. For MetaQA and PQL, we constructed an alternative N-Hop Agent with capabilities similar to our Traversal Agent but equipped with a different hop-based toolkit detailed in the Appendix B. In all our experimental setups, we used GPT-4.1-mini as our LLM and text-embedding-3-small as the embedding model.

## 4.1 Experiment Results

The performance of our models is summarized in Table 1 for recall and Table 2 for precision. Across the board, agent-based approaches demonstrate a clear advantage over traditional dense retrieval, especially on complex ontologies. In D-WIS, the Traversal Agent with Reflection achieved a recall of 92.5% and precision of 71.2%, significantly outperforming all baselines. A similar trend was observed in the GBO ontology, where reflection improved precision from 69.5% to 75.9%. These results underscore the effectiveness of dynamic traversal and reflective refinement in dense and structured environments.

However, the results on the FoF ontology highlight an important nuance: the diminishing returns of reflection when initial performance is already high.

The Transverse Agent achieved a high recall of 89.7% on FoF without reflection, and the reflection loop provided no further improvement. Moreover, precision saw a marginal decrease. This is because FoF is a simple graph, and the queries generated by the user simulator do not require deep, multi-hop traversal. The initial nodes retrieved are often sufficient and correct, leaving little room for a reflective process to add value. In such cases, the overhead of reflection can lead to redundant inefficient exploration rather than meaningful refinement.

On the established KG benchmarks, the Traversal Agent with Reflection consistently outperformed its non-reflective counterpart. In MetaQA, reflection increased the recall from 71.5% to 84.3% and in PQL from 85.4% to 88.1%, confirming that the reflection loop is highly effective for tasks that require navigating multiple hops and reasoning through complex paths.

## 4.2 Tool Use

To understand how reflection improves performance, we analyzed the agent's tool utilization patterns, as shown in Table 3. The data reveal a clear shift in strategy when the reflection mechanism is active, with the effect varying by the complexity of the data set.

On complex graphs such as D-WIS and GBO, reflection guides the agent toward a more efficient strategy. For D-WIS, the agent reduces its reliance on initial broad searches ('get_best_matches') and property checks ('get_predicates_for_entity'), indicating that it learns to trust its initial grounding and avoid redundant information gathering. Similarly, on GBO, the agent significantly reduces broad neighbor exploration (from 6.1 to 3.6 calls) in favor of a more targeted traversal. This pattern is also visible on MetaQA and PQL, where reflection encourages a more focused navigation path.

Conversely, on the simple FoF ontology, the reflection loop appears to induce overthinking. The reflection agent makes more calls to nearly every

**Table 4.** Ablation study results on the D-WIS dataset, highlighting the critical role of the reflection mechanism and the predicate-checking tool.

| Model | D-WIS Rec | D-WIS Prec |
|---|---|---|
| w/ Ref | **92.5 %** | **71.2 %** |
| w/o Ref | 90.8 % | 68.4 % |
| w/o Ref and Search | 89.0 % | 67.9 % |
| w/o Ref and Predicate | 84.7 % | 66.6 % |

tool, most notably increasing its 'get_neighbors' calls from 1.5 to 3.8 and 'get_predicates_for_entity' from 0.3 to 1.3. This aligns with the precision results and supports the hypothesis of diminishing returns: on a task that is already simple, the reflective process causes the agent to explore unnecessarily, leading to inefficiency without a corresponding performance gain. Overall, reflection successfully teaches the agent to be more deliberate, but its utility is directly proportional to the complexity of the traversal task.

## 5 Ablation Study

To further isolate the impact of specific components, we conducted an ablation study on the complex D-WIS ontology, with results presented in Table 4. The study systematically removes key tools and the reflection mechanism to quantify their contribution.

The full Traversal Agent with Reflection serves as our top-performing benchmark. Removing the reflection mechanism alone caused a drop in 90.8% recall and 68.4% precision, confirming that the reflection loop is a significant driver of performance. Removing the agent's ability to perform its own initial entity search ('get_best_matches') led to a further decrease, demonstrating the value of allowing the agent to dynamically ground its starting point in the graph.

Critically, the most substantial performance degradation occurred when we removed the tool 'get_predicates_for_entity'. Recall fell to 84.7%, and precision fell to 66.6%, which highlights that the agent's ability to inspect the types of relationship a node possesses is fundamental to its navigation strategy. Without being able to preview connections, the agent navigates blindly, leading to less effective and incomplete traversals. This underscores the importance of tools that provide structural awareness for efficient navigation in complex graphs.

## 6 Discussion

Our experimental results demonstrate that an autonomous agent with traversal tools and a reflective feedback mechanism is a highly effective approach to information retrieval in large and dense knowledge graphs. The findings confirm our central hypothesis:

for complex, nonprunable structures like the D-WIS and GBO ontologies, dynamic agent-based traversal significantly outperforms traditional, static retrieval methods.

The key to this success is the process-oriented reflective loop. By focusing on the use of tools, the agent learns to refine its exploration strategy. This is a crucial distinction from previous work such as Self-RAG, which critiques the retrieved content. Our method improves the retrieval process itself, leading to more efficient navigation, as seen in the tool-use analysis for D-WIS, GBO, and the multi-hop KG datasets.

However, our work also introduces a critical finding regarding the context-dependent utility of reflection. In the simple FoF ontology, where traversal paths are short and initial precision is high, the reflection mechanism offered no benefit and slightly increased inefficiency. This illustrates the principle of diminishing returns: reflection is most valuable when the task is complex, and there is significant room for improvement. For straightforward queries, the overhead of a reflection loop is unnecessary. This suggests that future intelligent systems could dynamically toggle reflection based on initial task assessment.

Our ablation study further solidifies the importance of a well-rounded tool set. Although semantic search provides a necessary entry point, the ability to inspect node predicates ('get_predicates_for_entity') proved to be the most critical. This indicates that awareness of the local graph topology is paramount for an agent to navigate intelligently.

While the agent shows strong recall and precision in retrieving relevant nodes, this does not directly guaranty the correctness of the final answer in a downstream QA or generation task. The current evaluation focuses on retrieval quality, and future work should assess end-to-end performance, including factual accuracy and coherence of generated responses.

Additionally, the scoring agent introduces a potential bias, as it relies on an LLM to evaluate the output of another LLM, both based on the same model. This may lead to inflated performance estimates due to shared reasoning patterns. A more robust evaluation would involve human judgment, diverse model ensembles to validate the retrieval quality, or changing the scoring to a relative scale with additional final decision to obtain the final outcome.

## Limitations

Although our findings demonstrate the promise of reflective agent-based traversal, the study has several limitations. First, our evaluation of ontology datasets is based on a user simulator. These sim-

ulated queries are syntactically and semantically idealized and may not reflect the ambiguity, noise, or variability of real-world user input. This could lead to an overestimation of performance.

Second, the scope of our evaluation is limited to five specific knowledge graphs. The effectiveness of our agent's strategies may not generalize to other graph types, such as sparse graphs, dynamic graphs, or social networks, which may require different tools or traversal heuristics.

Third, the scoring agent introduces bias by using an LLM to evaluate another LLM's output. This self-referential evaluation may reinforce model-specific reasoning patterns and lacks external validation.

Fourth, our study focuses exclusively on the retrieval phase, using precision and recall as primary metrics. We do not evaluate the quality of the generated natural language output, which limits insight into the system's end-to-end utility.

Fifth, while Table 3 provides tool usage statistics across datasets of varying complexity, it does not yet include detailed measurements of runtime, memory consumption, or cost per traversal iteration. These metrics are important for assessing the practical scalability of the approach, especially in large-scale or real-time applications. A more comprehensive profiling will be necessary to explore the scalability of the method.

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

# A Prompts

**Agent Prompt Tempale**

Listing 1. Prompt for Ontology Traversal

```
**You are an intelligent agent exploring an ontology graph.**

Your mission is to **identify relevant concepts** for:
'{search_terms}'

**Primary Goal**: Locate all nodes capturing the concepts related to the search
    terms (e.g.
Classes, Predicates, Properties).

You are limited to:

* **{max_moves} total moves** (to navigate the graph)
* **{max_marks} total marks** (to record relevant nodes)

### Tools Available:

{tools_section}

### Strategy Tips:

* Start with fuzzy search to locate candidate entities related to the search
    terms
* Use your moves wisely - the ontology is large, and random wandering is
    inefficient
* Explore dense or semantically promising regions of the graph
* To be able to use 'get_neighbors', you must be in a node (use 'move_to' to get
    there)
* You can jump directly to any URI if you suspect it's useful'

### Success Criteria:

-  **All relevant components are found in marked nodes**
-  **Multiple nodes may each contain a part of the user intent** - you must **
    mark them all**
-  **You do not mark irrelevant or unrelated nodes**
-  **Note:** If you want a seed node to be included in the final results, you **
    must mark it**
using 'mark_nodes'.

You will succeed if you:
- Collect a complete, relevant set of nodes, describing user intent using only
    the marked nodes
```

### Scoring Prompt Template

**Listing 2.** Prompt for Scoring Ontology Traversal with example Output

```
## Task
Score the agent's performance on *Relevant Query Node Marking* on a scale of
    0-100, based on task completeness and precision of the selected nodes.

## Initial User Queries
{user_queries}

## Final Agent State
**Marked Nodes:**
{marked_nodes}

## Scoring Criteria

### Task Completeness (Recall) - 70% weight
- **100%**: All relevant classes/predicates directly or semantically matching
    the user query are marked
- **75%**: Most relevant nodes marked; at most one important node missed
- **50%**: Some relevant nodes marked; several important ones missing
- **25%**: Few relevant nodes identified; large gaps in query coverage
- **0%**: No relevant nodes identified

### Information Redundancy (Precision) - 30% weight
- **100%**: All marked nodes are highly relevant; no unnecessary or incorrect
    markings
- **75%**: Most marked nodes are relevant; minor irrelevant selections present
- **50%**: Mixed relevance; some clearly irrelevant or unnecessary nodes
- **25%**: Many marked nodes are off-target or unrelated to the query
- **0%**: Markings are mostly incorrect or irrelevant
```

### Reflection Prompt Template

**Listing 3.** Prompt for Reflecting Traversal

```
## Task

Analyze the agent's tool usage patterns and provide actionable improvement
    suggestions based on the conversation trace and final performance score.

## Agent Conversation Trace
{messages_trace}

## Remaining Resources
- Moves: {remaining_moves}/{max_moves}
- Marks: {remaining_marks}/{max_marks}

## Final Performance Score
{final_score}/100

## Analysis Framework

### Tool Usage Patterns to Evaluate
1. **Strategic Tool Selection**: Did the agent choose the most effective tools
    for each situation?
2. **Resource Efficiency**: How well did the agent manage limited moves and
    marks?
3. **Information Gathering**: Was the exploration thorough enough to make
    informed decisions?
4. **Timing of Actions**: Were tools used in the optimal sequence?
5. **Redundancy vs. Completeness**: Did the agent balance efficiency with
    thoroughness?

### Tool-Specific Analysis Points
```

#### get_best_matches
- Was this used early enough to establish search direction?
- Were search terms well-formulated?
- Did the agent act appropriately on the results?

#### get_neighbors
- Was this used strategically or randomly?
- Did the agent explore promising neighborhoods?
- Was the information from neighbors utilized effectively?

#### move_to
- Were movements purposeful or wasteful?
- Did the agent visit the right nodes?
- Was the exploration pattern efficient?

#### mark_nodes
- Were marks used at the right time?
- Was the confidence assessment accurate?
- Did the agent balance exploration vs. marking?

#### end_traversal
- Was the termination decision well-timed?
- Did the agent leave resources unused that could have improved results?

#### get_predicates_for_entity
- Were predicates explored to understand relationships?
- Did the agent use predicates to connect nodes meaningfully?

## Example Reflection Structure
```
### Tool Usage Summary
The agent demonstrated efficient semantic search but showed limited exploratory
    behavior...

### Key Strengths
- Excellent use of get_best_matches to quickly identify relevant nodes
- Strategic resource management with 5/10 moves used effectively

### Primary Improvement Areas
- Insufficient exploration of node relationships and context
- Premature marking without thorough verification

### Tactical Recommendations
- Use get_neighbors on marked nodes to verify context completeness
- Consider exploring related measurement units before final marking

### Strategic Insights
- Balance efficiency with thoroughness when stakes are high
- Develop heuristics for when to explore vs. when to mark
```

Focus on actionable, specific improvements rather than general observations.

## User Simulator Prompt Template

**Listing 4.** User Simulator Prompt with example Output

```
**You are a simulated user interacting with a knowledge graph creation tool.**

Your task is to write a **clear, natural-sounding instruction** that either:

1. **Adds new knowledge** (e.g. inserting a fact about a person, event, place,
   etc.), or

You will be provided with a list of ontology terms (classes, properties). These
   terms represent concepts in the graph.

Your goal is to **formulate a realistic and natural instruction** that:
- Could plausibly come from a non-technical user interacting with a smart
   assistant or tool.
- Implies the provided ontology terms (Paraphrase or express them more naturally
   ).
- Makes logical sense in the context of building or querying a knowledge graph (
   e.g., "Add a new researcher who works at a university, then add that they
   are working on a project about AI.").

Avoid:
- Instructions that are vague, contradictory, or ungrounded in real-world
   concepts.
- Highly abstract or technical language (like "create a subclass of X" or "
   instantiate this axiom").

## Examples of good instructions:
- "Add a new city named Trondheim located in Norway. Then add a new landmark in
   the city."
- "I want to add a book titled 'The Hobbit' written by J.R.R. Tolkien. After
   that, add a new character named Bilbo Baggins."
- "Show me all employees who work at OpenAI."

## Important:
- If a term cannot be naturally used in a realistic sentence, you may skip it.
- Make sure that **all used terms** (even if paraphrased) are **listed as URIs**
    at the end.
- Do **not** list terms that were not actually used.
- You must use at least 3 ontology terms in your instruction.

Respond in the following format:

{
"instruction": "Your natural-sounding instruction here.",
"used_terms": ["<uri1>", "<uri2>", ...]
}

EXAMPLE OUTPUT:
{
"instruction": "Add a new wellbore record with detailed data about the formation
     strength encountered. Then include descriptions of the drill stem and the
   specific drill string used in the operation.",
"used_terms": ["http://ddhub.no/WellBoreData", "http://ddhub.no/
   FormationStrengthDescription", "http://ddhub.no/DrillStemDescription", "http
   ://ddhub.no/DrillStringDescription"]
}
```

# B  N-Hop Agent

N-Hop Agent is our alternative variant of Traversal Agent that is more tailored to specifically RDF triples. The ability to traverse the graph through get_neighbors and move_to is replaced by get_triples_by_subject_predicate and get_triples_by_predicate_object. This complements the get_predicates_for_entity tool as together they allow to obtain all available triples in Knowledge Graph. We have chosen to focus with this agent only on Question Answering, as we have observed that ontology traversal does not often require travel, with Traversal Agent using the move_to tool only to confirm it's presence.

**N-Hop Agent Prompt Template**

**Listing 5.** N-Hop Agent Prompt Template

```
**You are an intelligent agent performing n-hop graph traversal to answer a
    specific question using structured exploration of a knowledge graph.**

Your mission is to **systematically explore the graph** to locate and mark nodes
    that contain the information required to answer:
'{search_terms}'

**Primary Goal**: Identify and mark all entities that contribute to answering
    the question

> **Important**:
> You must mark **every node** that contains part of the answer.
> If a required answer component is not marked, the answer will be incomplete.
> e.g., for "What are the genres of films directed by [Person X]", both 'Drama'
    and 'Comedy' must be marked if relevant

You are limited to:
* **{max_marks} total marks** (to record answer-relevant nodes)

### N-Hop Tools Available:
{tools_section}

### N-Hop Strategy:

* **Start with fuzzy search** to locate candidate entities related to the
    question
* **Explore predicates** from those entities to navigate meaningful
    relationships
* **Use subject/predicate or predicate/object traversal** to find related nodes
    that may contain answers
* **Mark nodes immediately** if they contain full or partial answers
* **Use bidirectional exploration** where helpful - some answer components may
    be found by reversing direction

### Traversal Tips:

* Use 'get_predicates_for_entity' to discover properties of entities
* Use 'get_triples_by_subject_predicate' to follow specific relationships
    forward
* Use 'get_triples_by_predicate_object' to find entities pointing to a target
* **Don't assume a single node holds the full answer** - be alert to partial
    answers spread across multiple nodes
* **Mark all answer-contributing nodes, not just final answers**

---

Stay focused on **efficiently gathering all answer-critical information**. Your
    goal is not to explore the whole domain or construct reasoning chains - it's
    to **identify and mark all the pieces of information** required to fully
```

```
    and accurately answer the question.

---

### Success Criteria:

-   **All answer components are found in marked nodes**
-   **Multiple nodes may each contain a part of the answer** - you must **mark
    them all**
-   **You do not mark irrelevant or unrelated nodes**

You will succeed if you:
- Collect a complete, correct answer using only the marked nodes
```

# C  Examples

Here we would like to present generated analysis from both the Scoring and Reflection Agent on the PQL and MetaQA benchmarks.

## C.1  Scoring Example

The example chosen for the Scoring Agent is one in which the Traversal Agent scored 75 during his first attempt and 100 during his second attempt. The general problem that could be seen is that only partial answers are given. The example below Peter Hook was not given any instrument, because he does not have any associated in the ontology. The issue was that the Agent did not mark 'Peter_Hook' thus the associated properties are not visible to the scoring agent.

**Listing 6.** Example Scoring over insufficient answer

```
To evaluate the agent's performance based on the provided user query and the
    final agent state, we will analyze both task completeness and information
    redundancy.

### Task Completeness (Recall)
The user query is asking for the instruments played by the lyricist of the song
    \"Isolation.\" The agent identified two lyricists: Peter Hook and Ian Curtis
    . However, the query specifically asks for the instruments played by the
    lyricist, which is only relevant to Ian Curtis in this context, as he is the
     one whose instruments are listed.
1. **Identified Lyricists**:
    - Ian Curtis (instruments played: Keyboard, Guitar)
    - Peter Hook (no instruments mentioned)
2. **Instruments Played**:
    - The agent correctly identified Ian Curtis's instruments (Keyboard, Guitar)
        .
The agent successfully found the relevant information regarding Ian Curtis's
    instruments, but it also included Peter Hook, which is not necessary for
    answering the query. However, since the query is about the lyricist's
    instruments, and Ian Curtis's instruments were found, we can conclude that
    the agent has covered the essential part of the query.

**Score for Task Completeness**:
- The agent found the relevant node for Ian Curtis and his instruments, but
    included an unnecessary node for Peter Hook. Therefore, it can be rated as
    **75%** (found most relevant nodes, single node missed).

### Information Redundancy (Precision)
Now, we will evaluate the relevance of the marked nodes:
1. **Marked Nodes**:
    - \"Isolation\" (relevant)
    - \"Ian Curtis\" (relevant)
    - \"__music__group_member__instruments_played\" (relevant but redundant as
        it doesn't provide new information)
```

```
      - \"Keyboard\" (relevant)\n   - \"Guitar\" (relevant)
The agent included a node that indicates the instruments played by Ian Curtis,
    which is somewhat redundant since the instruments are already specified in
    the nodes for Keyboard and Guitar. However, all other nodes are relevant to
    the query.

**Score for Information Redundancy**:
- Since most marked nodes are relevant but there is some redundancy, the score
    can be rated as **75%** (most marked nodes are relevant, minimal irrelevant
    selections).

### Final Score Calculation
Now, we will calculate the final score using the weights provided:
- Task Completeness Score: 75% (0.75)
- Information Redundancy Score: 75% (0.75)

Final Score = (0.75 * 0.7) + (0.75 * 0.3) = 0.525 + 0.225 = 0.75
To convert this to a scale of 0-100, we multiply by 100:
Final Score = 0.75 * 100 = **75**

### Conclusion
The agent's performance is scored at **75** based on the criteria of task
    completeness and information redundancy.
```

## C.2   Reflection Example

As can be observed in the example below, the reflection agent properly identified the errors caused by not marking early and getting confused by the marks allowed. Reflection also observed problems with movement and repeated querying. In the next attempt, the model was able to improve, obtaining a score of 65 and returning more correct answer nodes.

**Listing 7.** Example Reflection over insufficient answer

```
SCORE: 50
### 1. Tool Usage Summary
The agent effectively initiated the traversal with 'get_best_matches' and used '
    get_predicates_for_entity' to understand entity relationships. However, the
    agent over-relied on repeated 'get_neighbors' calls on the same node without
     caching results, leading to inefficient move usage. Marking was done late
    and in bulk, causing the agent to exceed the allowed mark limit and
    resulting in errors. The traversal ended prematurely without fully
    leveraging remaining moves or marks to ensure completeness.

### 2. Key Strengths
- **Early and effective use of 'get_best_matches'**: Quickly identified the
    target film "Central Station" and its screenwriters, setting a clear
    exploration direction.
- **Predicate exploration to guide traversal**: Used 'get_predicates_for_entity'
     on key entities (film and screenwriters) to identify relevant relationships
     like 'written_by' and 'directed_by'.
- **Focused exploration of screenwriter neighborhoods**: Moved to and explored
    neighbors of "Walter Salles" to find related films sharing the same
    screenwriter, aligning well with the question's goal.

### 3. Primary Improvement Areas
- **Redundant 'get_neighbors' calls on the same node**: Multiple identical calls
     on "Walter Salles" yielded no new information, wasting moves and time.
- **Mark overuse and poor mark management**: Attempted to mark 6 nodes despite
    only having 5 marks, causing mark failures and confusion.
- **Delayed and bulk marking strategy**: Marking was postponed until many nodes
    were identified, increasing the risk of exceeding mark limits and losing
    track of confirmed answer components.

### 4. Tactical Recommendations
```

- **Cache neighbors from the first `get_neighbors` call per node** and reuse
  this data instead of repeating the same query.
- **Mark nodes incrementally as soon as they are confirmed answer components**
  to avoid exceeding mark limits and to maintain clarity on what has been
  confirmed.
- **Use `get_predicates_for_entity` to selectively guide neighbor exploration**
  rather than broad or repeated neighbor queries.
- **Leverage `move_to` strategically to explore new nodes rather than repeatedly
  querying neighbors from the same node**.
- **Track remaining marks and moves continuously** to avoid resource exhaustion
  and premature traversal termination.

### 5. Strategic Insights
- **Plan exploration paths that maximize new information per move and mark** by
  prioritizing predicates and relevant nodes.
- **Adopt a progressive marking strategy** to ensure completeness and avoid
  errors related to mark limits.
- **Use predicates as a roadmap for n-hop traversal** to focus on meaningful
  relationships rather than broad neighbor exploration.
- **Continuously monitor resource usage and adapt exploration depth or breadth
  accordingly** to balance thoroughness with efficiency.
- **Avoid redundant queries by maintaining internal state or caches** to improve
  resource efficiency and traversal speed.
---
By implementing these improvements, the agent can achieve more efficient,
accurate, and complete graph traversals within the given resource
constraints, leading to higher performance scores and more reliable answers.

