# OpenReview forum: "Reflective Agents for Knowledge Graph Traversal"
_NLDL.org/2026/Conference — NLDL 2026 Poster_

### Official Review · Reviewer_KPc5 · 2025-10-07
**Reflective Agents for Knowledge Graph Traversal: Review**

**Rating:** 4
**Confidence:** 4

**Summary:**

The paper Reflective Agents for Knowledge Graph Traversal introduces a new way to implement RAG on Knowledge Graphs by combining a feedback-optimised traversal of the graph, trying structural information, reflection and agent-based models. The novelty lies in optimising the traversal instead of retrieved information, to better navigate the graph and increase performance. Experiments show that the framework works well on complex ontologies, but over-complexifies when the relationships are simple (e.g. FoFs). It opens the door to avoid pruning on complex ontologies.

**Strengths:**

- Strong and principled combination of existing literature, leading to a simple new method which takes the best aspects of popular existing methods
- Experimental results clearly demonstrate that Reflective Agents outperform current models (Table 1) and have an interpretable mechanism (Table 3)

**Weaknesses:**

- Table 3 somewhat undermines the scalability and efficiency claims. We see at most a decrease of 3 tool calls and at worst no decrease at all. Perhaps different markers of efficiency and scalability could be shown, besides comparative performance. Even if deferred to the appendix, an explanation or heuristic on the efficiency gain (e.g. asymptotics) would strengthen the claim in my eyes.
- Figure 1 does not reflect the fact that the scoring agent knows the use query, but not the reflection agent. The mechanisms are not very clearly explained.

**Justification:**

This paper introduces the intuitive notion of combining adaptive graph traversal, which is both principled and works. This makes a contribution to the field by combining existing techniques, which only requires the use of widespread tools. However, the explanation beyond heuristics is a little lacking, at the risk of overselling the efficiency and scalability. I do not believe that new experiments are needed; the results speak for themselves but the presentation and explanations could use some work (comment on Table 3 and Figure 1).

---

> ### Author Rebuttal · Authors · 2025-10-15
>
> We thank the reviewer for the positive evaluation and the excellent suggestions for improving the paper's clarity.
>
> - On Efficiency Claims and Table 3: This is a great point. The efficiency gain is clearer when considering that tools have different computational costs. `move_to` and `mark_node` calls, returning single node info, are much less complex compare to `get_best_matches` and `get_neighbors` which return large list of nodes. For the most complex Datasets (D-WIS, GBO, MetaQA) we see decrease in usage of  `get_best_matches`.  D-WIS have larger amount of `get_neighbors` calls with reflection, which undermines the efficiency, but is traded for better recall and precision.
> - On the Clarity of Figure 1: The reviewer is correct that the figure is ambiguous. We argue that the caption of the figure describes the process enough, but we can accept that the figure does not describe the process in optimal way.
>
> Thank you again for the highly constructive feedback.

---

### Official Review · Reviewer_wVcL · 2025-10-08

**Rating:** 4
**Confidence:** 2

**Summary:**

The paper, "Reflective Agents for Knowledge Graph Traversal," tackles a key challenge in the field of Retrieval Augmented Generation (RAG): how to effectively pull information from dense, highly structured databases like knowledge graphs. Traditional RAG methods often struggle with the scale and complexity of these graphs.

**Strengths:**

1. The results, as presented in the paper, are impressive. The Reflective Agent with its feedback loop consistently outperforms baseline methods, particularly on complex graphs. The paper also highlights a significant increase in recall for the MetaQA and PQL datasets when reflection is enabled.

2. One of the paper's key contributions is its detailed analysis of the agent's "tool utilization."

**Weaknesses:**

1. The reflective mechanism is not always beneficial, the marginal effects of simple knowledge graphs need to be considered.

2. The entire reflection mechanism hinges on the feedback provided by the "Scoring Agent." The paper provides no validation for this critical component.

**Justification:**

I don't think the paper can guarantee that the scores of the grading agent are accurate, this LLM-as-a-judge approach is susceptible to the same biases

---

> ### Author Rebuttal · Authors · 2025-10-15
>
> We sincerely thank the reviewer for the positive assessment and for raising two very insightful points.
>
> - On Reflection's Diminishing Returns: We agree. We believe it is an important finding of our paper that reflection can cause the agent to overthink simple tasks. In our opinion great research about role of context can be found in paper "Large Language Models Can Be Easily Distracted by Irrelevant Context", supporting this claim.
> - On the Scoring Agent: The reviewer makes a fair point. Our approach is similar to that of Self-RAG, where the goal is to rank multiple retrievals. The value of the combined reflection mechanism is demonstrated in our ablation study, which shows a performance gain. We agree that the scoring could be improved, for example making the scoring more about the belief to improve (-2 to +2), with final ranking of all retrievals.
>
> Thank you again for the thoughtful and constructive feedback.

---

### Official Review · Reviewer_G4fr · 2025-10-08

**Rating:** 2
**Confidence:** 2
**Final Rating:** 2
**Final Confidence:** 3

**Summary:**

This paper introduces a "Reflective Agent" to improve information retrieval from large, dense knowledge graphs where traditional methods like graph pruning are ineffective. The autonomous agent dynamically traverses the graph using a set of tools. The core innovation is a self-reflection loop that analyzes the agent's tool usage and performance, providing feedback to optimize its navigation strategy.

Experiments on various ontologies and datasets show that this agent-based approach significantly outperforms standard retrieval methods. The reflection mechanism enhances traversal efficiency and accuracy on complex graphs, though its benefits diminish on simpler tasks.

**Strengths:**

- It proposes a new synthesis of autonomous agents, graph traversal, and a self-reflection mechanism focused on process optimization rather than content critique.

- The paper provides an analysis of the agent's behavior through a detailed examination of its tool utilization patterns, explaining how reflection improves performance.

- The authors demonstrate strong scientific practice by including a clear and honest "Limitations" section, acknowledging potential weaknesses in their study.

- The paper promotes reproducibility by providing the full prompts for the traversal, scoring, reflection, and user simulator agents in the appendix.

**Weaknesses:**

- The evaluation is fundamentally disconnected from the end goal. The paper measures the precision and recall of retrieved nodes but completely fails to evaluate the quality of the final, generated natural language answer. A RAG system's success is ultimately judged by its output to the user, not its intermediate retrieval performance. It's entirely possible to have high retrieval recall but still generate a nonsensical or incorrect answer, a critical failure mode this paper does not address.

- The use of a "User Simulator" creates an artificial and overly sanitized test environment. Real-world user queries are ambiguous, contain misspellings, use colloquialisms, and often lack specific entity names. The simulator, which is prompted to create "clear, natural-sounding instructions" based on known ontology terms, generates perfect-world problems that are likely far easier than what the agent would face in a real application. This calls the impressive performance metrics into serious question.

- The claims of handling "large-scale" graphs are not convincingly supported. The primary "complex" ontology, D-WIS, has around 6,000 classes. While not trivial, this is orders of magnitude smaller than truly large, web-scale knowledge graphs with millions or billions of nodes and edges. The iterative, multi-agent LLM call structure would be prohibitively expensive and slow at that scale, a critical scalability issue that is glossed over.

- The evaluation methodology is inherently subjective and self-referential. Using one LLM (the Scoring Agent) to grade the performance of another LLM (the Traversal Agent) introduces significant potential for bias, inconsistency, and unreliability. The scoring is based on the LLM's interpretation of "relevance," not on a deterministic, objective ground truth, making the quantitative results less robust than they appear.

**Final Justification:**

My core concerns (see below) are not actually addressed by the authors' rebuttal. I still lean towards rejection.

1. The evaluation is fundamentally incomplete. The study measures the precision and recall of retrieved nodes but completely fails to evaluate the quality of the final generated answer.

2. The experimental setup lacks real-world validity. The use of a "user simulator" to generate clean, well-defined tasks creates an artificial environment.

3. The claims of scalability are unsubstantiated.

**Justification:**

While the paper tackles an important problem, I feel there are still some significant methodological flaws that undermine its core contributions:

1. The evaluation is fundamentally incomplete. The study measures the precision and recall of retrieved nodes but completely fails to evaluate the quality of the final generated answer.

2. The experimental setup lacks real-world validity. The use of a "user simulator" to generate clean, well-defined tasks creates an artificial environment.

3. The claims of scalability are unsubstantiated.

Therefore, I lean towards rejection.

---

> ### Author Rebuttal · Authors · 2025-10-15
>
> We thank the reviewer for their constructive feedback and appreciate the opportunity to clarify the following points:
>
> - On Evaluation Scope: Our paper's contribution is focused on the retrieval stage, a critical and self-contained challenge for dense Knowledge Graphs where standard methods are ineffective. While end-to-end generation is important future work, improving this retrieval bottleneck is a valid and necessary step.
> - On the User Simulator: The simulator ensures a fair, controlled comparison between our method and the baselines. To validate these results, we also tested on the standard MetaQA and PQL benchmarks, which use real questions and showed our approach’s gains are consistent.
> - On Graph Scale: We want to clarify the graph complexity. The D-WIS ontology is altogether around 13k axioms (Taking 200k Tokens), and MetaQA contains around 135k triples (1.5M tokens). Our method targets such large, domain-specific graphs that are too big for in-context learning, not necessarily web-scale graphs, which we acknowledge as a limitation.
> - On LLM-based Scoring: LLM is deemed as uniquely qualified to capture and score if the underlying instruction/question can be satisfied by set of document. This approach follows work done in Self-RAG, for tasks that require scoring or ranking of retrieved context to implement the ability to reflect.
>
> We hope these clarifications help address the reviewer's concerns. Thank you again for your valuable feedback.

---

### Meta-Review · Area_Chair_cMTB · 2025-10-31

**Recommendation:** Accept (Poster)
**Confidence:** 3

**Metareview:**

This paper combines 3 methods for knowledge graph retrieval. The combination improves recall score of retrieval significantly. The introduction of a reflection agent to improve the traversal contributes to the performance increase.

The reviewers agree that the approach chosen combines existing work in an interesting way, showing performance improvements.
There are however a number of methodological weaknesses pointed by the reviewers:
- As acknowledged by the authors in the "Limitations" section, the "User simulator" will generate too "perfect" queries, which are likely to be unrealistically easy.
- The performance gain provide a limited insight on overall performance of the system to provide a correct answer,
- The method relies on a "Scoring agent" which is introducing a bias (LLM giving score to another LLM, both based on the same model), and is lacking a more thorough validation.
- The scalability claim is insufficiently backed. The experimental results presented in table 3 are not conclusive.

Under the assumption that the authors will clarify the limitations pointed by the reviewers in their final version, I would recommend the paper to be accepted.

---

### Decision · Program_Chairs · 2025-11-05

**Decision:**

Accept (Poster)

**Comment:**

We recommend a poster presentation given the AC and reviewers recommendations.